# *Mycobacterium marinum*: A Challenging Cause of Protracted Tenosynovitis

**DOI:** 10.3390/antibiotics12030629

**Published:** 2023-03-22

**Authors:** Pernille Grand Moestrup, Maiken Stilling, Christian Morberg Wejse, Victor Naestholt Dahl

**Affiliations:** 1Department of Infectious Diseases, Aarhus University Hospital, DK-8200 Aarhus N, Denmark; 2Department of Orthopaedics, Aarhus University Hospital, DK-8200 Aarhus N, Denmark

**Keywords:** nontuberculous mycobacteria, *Mycobacterium marinum*, tenosynovitis, diagnostic delay, clinical management

## Abstract

*Mycobacterium marinum* infections are rare, and they can be difficult to diagnose and treat. This may lead to further spread of the infection and complications, such as tenosynovitis, pyomyositis, and osteomyelitis. A 40-year-old previously healthy man presented with tenosynovitis of the extensor tendons on the second phalanx of his right hand. He was initially treated with steroid injections without any effect. Followingly, ulceration and an abscess developed on the dorsal site of the hand. At this point, it came to the physician’s knowledge that the patient had been cleaning an aquarium before onset of symptoms. After progression to massive tenosynovitis, the patient was admitted and underwent multiple surgical debridements. Briefly, after the first surgery, an interferon-γ release assay was positive, and treatment for *M. marinum* with rifampicin and azithromycin was initiated after eight months of symptoms. Later, a surgical biopsy showed acid-fast bacilli, and a polymerase chain reaction confirmed the diagnosis of *M. marinum*. In this case story, we highlight the difficulties of diagnosing and managing this complicated infection, describe the considerable morbidity associated with it, and suggest that local tissue concentrations could be useful to improve clinical outcomes, as these concentrations are potentially suboptimal.

## 1. Introduction

*Mycobacterium marinum* infections are rare, with annual incidence rates ranging from 0.4–1.3 per 1,000,000 individuals [1,2]. Still, the infections may lead to serious, treatment refractory skin and soft tissue lesions [1,2]. *M. marinum* are slowly growing acid-fast rods that grows at temperatures below 37 °C, belonging to the nontuberculous mycobacteria (NTM) species [3]. These bacteria are ubiquitous in aquatic environments and are most often transmitted through contact with fresh and salt water, including marine organisms, swimming pools, and aquariums [4]. Consequently, these infections are known as “fish tank syndrome/granuloma”, “fish handler’s disease” or “fish tank finger” [5]. After traumas or exposure of skin lesions, the infection is typically directly inoculated [6]. Most commonly, infections result in slowly developing cutaneous manifestations as solitary papules or nodules (up to 92.5%), while only a few percentages cause tenosynovitis [2]. A wide range of microorganisms may lead to infection after exposure to water (mostly polymicrobial and Gram-negative bacteria), and this is also dependent on the type of water, geography, as well as individual patient factors [7].

The incubation period of *M. marinum* is often around three weeks, but it may last up to nine months [8]. The long incubation period, as typically seen for slowly growing acid-fast NTM, and the awareness of these bacteria, may blur the association between exposure and symptom onset, leading to a delay in the diagnosis. *M. marinum* is detected using microscopy, polymerase chain reaction (PCR), and culturing using techniques that are specific for mycobacteria [6]. Since a delay in diagnosis and specific antimicrobial treatment against *M. marinum* may be related to a higher risk of complicated infections, with spread to deeper tissues as joints and tendons, an early diagnosis is of utmost importance [9].

Here, we highlight the difficulties of diagnosing and managing this complicated infection in a patient with a protracted *M. marinum*-associated tenosynovitis after cleaning his aquarium. Initially, he was treated for degenerative tenosynovitis with steroid injections, but after eight months of symptom progression, the patient was finally diagnosed. The patient was treated successfully with extensive surgery with repeated infection debridement and antibiotics for six months.

## 2. Case Presentation

A 40-year-old man without any comorbidities, besides a previous gastric bypass surgery, was referred to the department of infectious diseases from the department of rheumatology on suspicion of tenosynovitis of an infectious origin. A timeline is shown in Figure 1.

Two weeks after the symptoms began with the finger being swollen, tender, and warm (Figure 2), the patient had a normal X-ray of the right hand taken at the emergency department. Biochemically, C-reactive protein was normal. He was referred to a private rheumatologist, and three months later, tenosynovitis of the extensor tendons of right hand was confirmed with magnetic resonance imaging (MRI). For six months, he received monthly steroid injections, while the dorsum of the patient’s hand had been increasingly swollen, with pain located around the second metacarpophalangeal joint. Gradually, an abscess developed proximal to the second metacarpophalangeal joint of his right hand (Figure 2B,C). When asked, the patient said he had been cleaning his aquarium without gloves before the onset of symptoms. He also had an extensive travel history and had hunting and scuba-diving as hobbies. Besides the skin lesion with redness and swelling of the dorsum of the hand and swollen axillary lymph nodes ipsilaterally, there were no distinctive findings during physical examination.

Four months after symptom onset, blood samples were still normal, including HLA-B27, RF-IgM, and anti-CCP (Table 1). Seven months after symptom onset, the abscess on the dorsum of the hand spontaneously ruptured (Figure 2C), and *S. aureus* was cultured from the wound. No acid-fast bacilli were found in the pus, and PCR did not show *Mycobacterium tuberculosis* complex DNA. At this point, the patient was referred to the department of infectious diseases.

A new MRI, nearly eight months after symptom onset, showed severe progression of the tenosynovitis, involving, at this point, all extensor tendons of the dorsal side of the right hand with extensive fluid accumulation in the distended hypertrophic synovium (Figure 3). Bone marrow oedema was seen in all carpal bones and in the base of the second metacarpal bone. In addition, fistulation to the levels below the extensor tendons was described. Due to the draining abscess and the massive tenosynovitis, the patient was admitted. Extensive surgical debridement of the infection, necrotic tissue, fistulation, and tenosynovitis was performed in sequential surgeries, which exposed extensor tendon thinning, stringing, and threatening rupture of the indicis proprium tendon. All extensor tendons were surrounded by a firm and immensely thick tendon sheath spreading dorsally from the base of the second finger and proximal to the middle of the forearm. No cutaneous manifestations in terms of nodules or papules were seen. An inoculation of tendon was sent for mycobacterial and pathological examinations. The next day, an interferon-γ release assay test came back positive, and the patient was started on antimicrobial treatment for *M. marinum.* Following this, microscopy showed acid-fast bacilli, and a PCR confirmed the tentative diagnosis. Histology showed acute and chronic inflammation with non-necrotic granulomas. However, mycobacterial culture later came back negative.

In the beginning, the patient was considered to have a degenerative tenosynovitis and was treated with steroid injections. At the department of rheumatology, there were initially no signs of inflammation at ultrasonographic examination leading to a suspicion of autoimmune disease, foreign body, or a slow infection. When the patient was referred to the department of infectious diseases, the suspicion of *M. marinum* was initially small, as the patient did not recall having any cutaneous lesions that could have been the bacterial site of entry. *Fransicella tularensis* was ruled out. However, over time, with the progression from swelling of one finger to the whole dorsum of the hand, the ulceration and abscess formation over the head of the second metacarpal, and especially the new information about scuba-diving activities and aquarium hobbies, the diagnosis of *M. marinum* infection seemed more and more obvious. Since *S. aureus* was cultured from the abscess, a second-generation cephalosporin (Cefuroxime, Braun) was administered intravenously at 1500 mg three times daily for one week after surgery without any improvement in clinical appearance of the infection site. Consequently, *S. aureus* was considered not to be the cause of disease.

The patient was admitted for 10 days and had three rounds of surgical debridement before being discharged (Figure 4). One month later, he had two more rounds of surgical debridement in the out-patient clinic. Pharmacologically, he was treated with rifampicin 600 mg and azithromycin 500 mg tablets daily. Low plasma concentrations of both azithromycin and rifampicin were later found. Rifampicin was gradually increased to 1200 mg and later 1800 mg. Azithromycin was also increased to 1000 mg, but it was again lowered due to high plasma concentrations and diarrhea. Prolonging the treatment because of the severity of the infection was considered, but the patient wished to cease treatment after six months because of adverse effects (mainly diarrhea and fatigue), which was considered clinically justifiable. The patient was seen regularly by an occupational therapist to retrain function of his hand. Since then, the wound has closed and healed without any major complications. The patient is regaining function of his hand, and the sensibility is improving. However, the patient is still suffering from scarring, redness of the hand, and pain on functioning of the hand (Figure 2D).

## 3. Discussion

Here, we presented a case of massive infectious tenosynovitis of the hand caused by *M. marinum.* The case demonstrates that the diagnosis is not easily made, and the treatment is complex and requires a multidisciplinary approach between medical and surgical specialists, as well as occupational therapists. Moreover, the case shows that these infections may be associated with a considerable morbidity (hospital admission, many out-patient visits, high burden of pills and extensive surgery, scarring, pain, and loss of function).

Among NTM, *M. marinum* most often cause skin lesions and have four predominant clinical manifestations: (1) superficial cutaneous lesions, (2) more than three lesions with a spreading pattern resembling sporotrichosis or abscesses and granulomas, (3) deeper infections, such as tenosynovitis (with or without cutaneous lesions), and (4) disseminated infections [2,10]. As shown by multiple studies, a vast majority *M. marinum* infections are seen in the upper extremity, involving hands and fingers (88.7–95%) [1,2,11], which is likely due to the higher risk of lesion exposure (handling of aquarium, fish etc.) and the bacteria’s grow at low temperatures. The infection is, in most cases, associated with an aquatic exposure (>80%) [2,11,12]. Tenosynovitis caused by *M. marinum* is commonly seen specifically in the hand or wrist because of the higher risk of penetrating injury [13]. One study found that fish tank exposure more often led to cutaneous manifestations, while boating and fishing were associated with invasive infections to higher degree, possibly due to deeper puncturing wounds [11]. Yet, the patient in our case did not recall any skin lesions prior to the swelling of the hand. Possibly, a lesion may have been forgotten due to the previously mentioned long incubation time of up to nine months and the potential diagnostic delay. In line with our case, steroid injections prior to the diagnosis are more commonly seen in invasive infections (29%) [11]. A corticosteroid-induced decrease in phagocytic activity of macrophages, reduced rate of infected cell death, and higher percentages of extracellular bacteria may explain this [14]. This emphasizes the importance of a thorough anamnesis and an early diagnosis to avoid a worsening of the infection [11].

The diagnostic delay for *M. marinum* may be extensive. One study found that the median time between the first symptoms and diagnosis was 6.5 months [2]. In our case, it was even longer (eight months). The delay may be caused by the rarity of the infections, but it may also be due to patient’s delay (i.e., time from symptom onset to contact with a health care provider) and health care delay (i.e., time from first contact with a health care provider to final diagnosis). Moreover, as suggested by Holden et al., the infection may mimic other more common skin and soft tissue infections, such as *S. aureus*. In line with this, *S. aureus* was cultured from the first inoculation of the abscess for the patient in our case. This could either be a contamination, colonization, or a disease-causing superinfection. However, the lesion did not improve when the patient was given cefuroxime intravenously. A similar case reported that suspicion of *S. aureus* or streptococci led to delay in diagnosis of *M. marinum* infection [15]. First of all, the clinician’s consideration of the specific infection is crucial to avoid delay. Furthermore, the low sensitivity and the time-consuming process of the diagnostic tests are considered, as a culture of *M. marinum* usually takes two to three weeks [5,6], which also delays the process. In only 13–16% of patients with a positive culture, acid-fast bacilli are found [2,11]. Only small studies are available on the sensitivity of PCR and culture with clinical cases as reference, showing that only 37.9% (*n* = 11/29) and 5.6–37.9% (*n* = 1/18 and 11/29) of patients had a positive PCR and culture [5,16], respectively. The diagnosis of *M. marinum* infection is based on symptomatology with characteristic lesions and a relevant history of marine exposure, histopathology showing granulomas, microscopy, PCR, culturing proving mycobacteria, and on a response to appropriate treatment [5,16]. If material is not sent for mycobacterial examinations at first, histopathology may be the first diagnostic clue.

Other diagnostic tools, such as immunohistochemistry, can be helpful. A study found that 34.5% (*n* = 10/29) were immunohistochemical positive, using a polyclonal antibody against *Mycobacterium bovis* with known cross-reactivity with a variety of mycobacteria, despite eight being culture-negative [16]. Another test that may be useful is the interferon-γ release assay that measures T-cell release of interferon-γ following stimulation of antigens specific to *Mycobacterium tuberculosis* (ESAT-6, CFP-10 and TB7.7) [17]. Consequently, a positive test may be indicative of *M. tuberculosis* exposure. Yet, these antigens are also shared by a few NTM species (*M. gastri*, *M. kansasii*, *M. riyadhense*, *M. szulgai*, and *M. marinum*), potentially leading to a positive result among some patients with NTM infection, as in our case [17]. The clinical manifestation in our case was suggestive of *M. marinum*, which is why the patient was started on treatment without a definitive diagnosis.

The evidence for treating of *M. marinum* infections is based on observational studies and in vitro studies showing susceptibility to rifamycins, macrolides, sulphonamides, trimethoprim, and ethambutol, but showing isoniazid and pyrazinamide resistance [6]. For superficial papules, monotherapy may be achievable, but for the deeper, more extensive infections, such as in our case, a reasonable suggestion is the combination of two active agents, typically clarithromycin and rifampin or ethambutol [6]. Although based on a low level of evidence, treatment is usually continued one to two months after resolution of symptoms totaling at least three to four months of treatment [6]. Routine susceptibility testing of isolates is not recommended for *M. marinum*, as there are no implications of significant mutational drug resistance and no substantial variation of susceptibility. However, there have been isolates with ethambutol, minocycline, and fluoroquinolones resistance [9].

One study found that deeper infections require considerably more surgical procedures (91.7%) compared with cutaneous *M. marinum* infections (23.5%) [9]. For infections involving closed spaces of the hand, surgical exploration and debridement is necessary to uncover the extent and involvement of the infection and to obtain tissue samples for cultures [11]. If the diagnosis is delayed, NTM infections may potentially penetrate deeper tissues, resulting in complications, such as tenosynovitis, pyomyositis, and osteomyelitis, occurring in up to one-third of cases [13]. More knowledge about local tissue concentrations of relevant antibiotics, as single drugs and in combinations, could be highly useful to understand and improve clinical outcomes, as these concentrations may be suboptimal. In our case, we even found low plasma concentrations. This was probably due to a high body weight (140 kg), as well as the patient’s previous gastric by-pass operation, but also due to the relatively low initial dosing of rifampicin. Historically, 600 mg of rifampicin has been considered the standard (maximum) dosage for tuberculosis [18], but it has become clear that it may more often be associated with inadequate exposures and unfavorable outcomes [19]. Suboptimal dosing of rifampicin may be plausible in slow response to therapy, failure, relapse, or acquired drug resistance, as rifampicin has a dose-dependent mycobacterial killing [20], emphasizing the usefulness of therapeutic drug monitoring [21] whenever available.

The long course to obtain the correct diagnosis and finalize treatment may also inflict other significant burdens on the patient. The whole course of disease typically results in numerous health care contacts and months of antibiotic treatment [1]. The consequences of repeated surgery, pain, and immobilization further lead to hospitalization and a long intensive rehabilitation to retrain hand function in expectance of some loss. Last, but not least, sick leave and economic and social consequences of a long diagnostic and treatment course is a significant burden for the patients.

In conclusion, *M. marinum* may cause complicated infections often with significant morbidity. The management is challenged by suboptimal diagnostic tools, delays in the diagnosis, a protracted effect of antimicrobial treatment, and a possible need of repeated surgery, which all together entails a risk of loss of hand function and lasting disability. More evidence is needed, and knowledge about local tissue concentrations could potentially be useful. Clinicians should be aware of *M. marinum* as a cause of slow and atypical skin and soft tissue infection. Possible aquatic exposures should be clarified initially, as they likely aid in a more direct diagnostic path. More research is warranted to improve diagnostics and treatment routes for *M. marinum* infections.

## Figures and Tables

**Figure 1 antibiotics-12-00629-f001:**
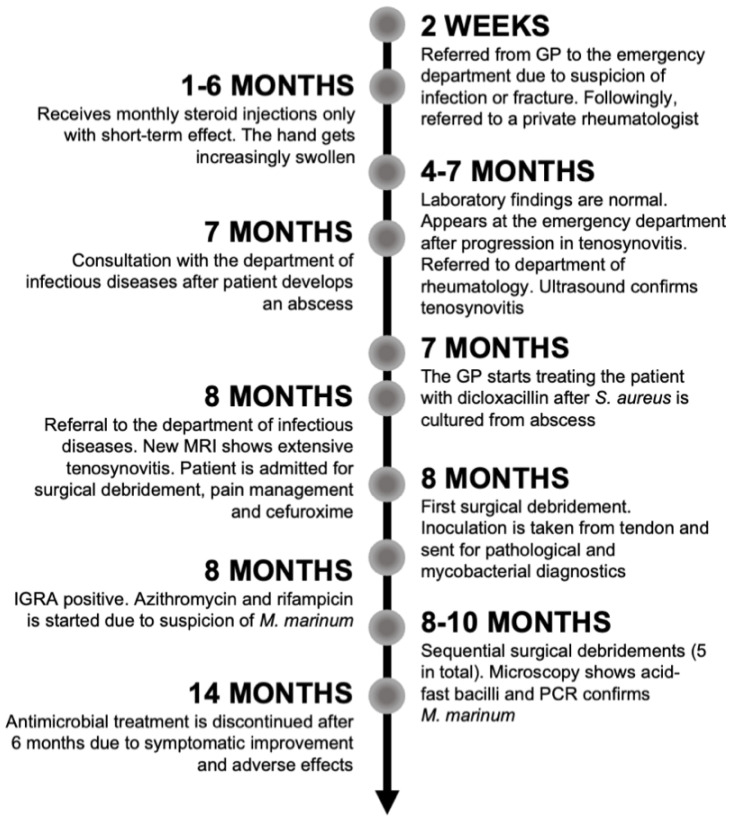
Timeline of the disease course after onset of symptoms. Abbreviations: GP, general practitioner. MRI, magnetic resonance imaging. IGRA, interferon-γ release assay. PCR, polymerase chain reaction.

**Figure 2 antibiotics-12-00629-f002:**
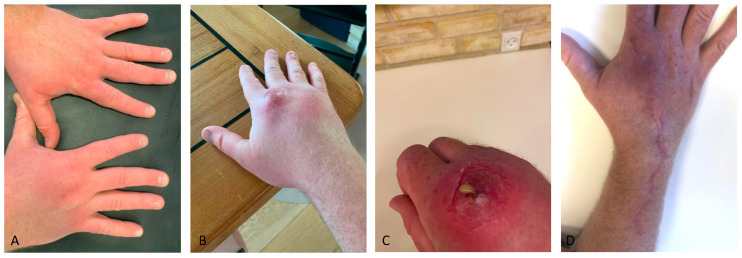
Appearance of the skin of the hand during the course of disease; (**A**): Swelling of the right index finger extending from the proximal phalanx to the dorsum of the hand over the second metacarpophalangeal joint (Week 10 after symptom onset). (**B**): Formation of an abscess (Week 27). (**C**): Spontaneous drainage of the abscess (Week 29). (**D**): After cessation of antibiotic treatment (Week 58).

**Figure 3 antibiotics-12-00629-f003:**
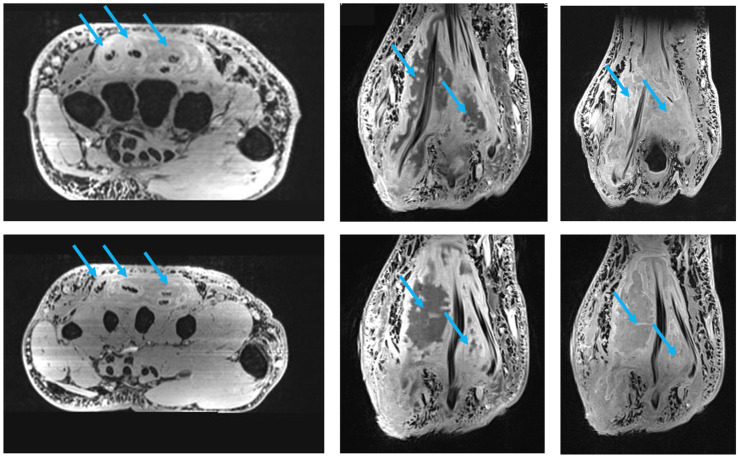
T1 sequence MRI images showing the massive tenosynovitis (blue arrows) in relation to the extensor tendons on the dorsum of the hand and wrist.

**Figure 4 antibiotics-12-00629-f004:**
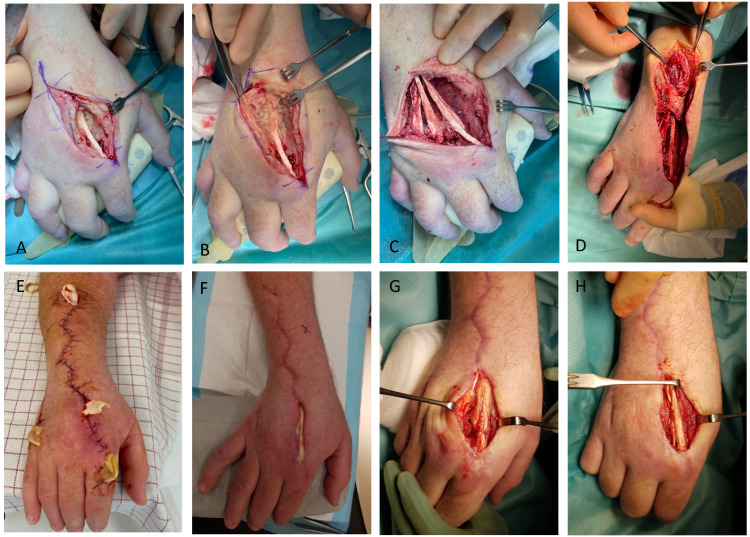
The appearance of the *M. marinum* tenosynovitis during the surgical treatment course. (**A**): The extensor tendons to the index finger have been debrided for a several millimetre thick firm tenosynovitis. (**B**): The sticky and firm tenosynovitis adhere to the tendon. (**C**): The extensor digitorum tendons 2 through 5 are debrided of the infectious tenosynovitis. The tendons are soft are and threading. (**D**): Second surgery, where it was found that the infection extended under the extensor retinaculum and proximal on the forearm along the tendons and muscle bellies. (**E**): Drains placed under the surgical scar. (**F**): The skin is healed, but the area over the tendons to the index finger remains open. (**G**,**H**): Third surgery, with extensor tenon exposure and debridement of more tenosynovitis.

**Table 1 antibiotics-12-00629-t001:** Biochemistry four months after symptom onset before referral to the department of infectious diseases. All analyses were within the normal range.

Analysis	Value	Reference
Alanine transaminase, U/L	54	10–70
Alkaline phosphatase, U/L	95	35–105
Glucose, mmol/L	7.1	4.2–7.8
Haemoglobin A1c, mmol/mol	43	<48
Thyrotropin, 10^−3^ IU/L	2.58	0.3–4.5
25-hydroxyvitamin D, nmol/L	102	50–160
Calcium, mmol/L	2.29	2.20–2.55
Potassium, mmol/L	3.9	3.5–4.6
Sodium, mmol/L	138	137–145
Albumin, g/L	41	36–45
Creatinine, µmol/L	85	60–105
eGFR/1.73 m^2^, mL/min	>90	>60
C-reactive protein, mg/L	<4.0	<8.0
Sedimentation reaction, mm	9	<15
Immunoglobin A, g/L	2.06	0.80–3.90
Immunoglobin G, g/L	7.9	6.1–14.9
Immunoglobin M, g/L	1.32	0.39–2.08
Leucocytes, 10^9^/L	6.8	3.5–10.0
Neutrophile granulocytes, 10^9^/L	3.86	2.0–7.0
Lymphocytes, 10^9^/L	2.18	1.30–3.50
Monocytes, 10^9^/L	0.59	0.20–0.70
Eosinophilocytes, 10^9^/L	0.13	<0.50
Basophilocytes, 10^9^/L	0.02	<0.10
Haemoglobin, mmol/L	8.5	8.3–10.5
Thrombocytes	247	145–350
Reumafactor-IgM, 10^3^ IU/L	1.5	<5
Cyclic citrullinated peptide antibodies, 10^3^ IU/L	2	<10
HLA-B27-gene	Negative	
Urate, mmol/L	0.34	0.23–0.48

Abbreviations: IU, international units. eGFR, estimated glomerular filtration rate. HLA, human leukocyte antigen.

## Data Availability

The data are not publicly available due to patient confidentiality.

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
