# Peer review of "Mycobacterium marinum*: A Challenging Cause of Protracted Tenosynovitis"

_antibiotics, 2023, doi:10.3390/antibiotics12030629_

Round 1
Reviewer 1 Report
Thank you for the opportunity to revise this interesting and timely manuscript. Extrapulmonary non-tuberculous mycobacteria (NTM) disease is an emerging clinical entity and may cause significant treatment challenges. The present manuscript describes these challenges and possible solutions which is of importance to clinicians handling such cases.
Major concerns:
1. In the introduction, please add information on the reason for M marinum location in extremities (grows at lower temperatures than 37 degrees Celcius)- This is also a reason for M marinums difficulty to grow in mycobatcerial blood cultures which is very rarely seen. Pleas also describe that it belongs to slow-growing NTMs in mycobacterial cultures which adds to diagnostic delay. An additional reason for diagnostic delay is lack of awareness regarding the possibility of NTM infection withous appropriate material sent for mycobacterial culture and histopathology. The latter can be stressed even more in the manuscript as histopathology may contribute to the suspicion of infectious granulomatous disease.
2. The following sentence needs to be questioned. "In our case, even plasma concentrations were low at first although standard doses were given.". A flat dose of 600 mg for a patient weighing 140 kg can hardly be considered standard. The recommended dose for rifampicin is 10 mg/kg and the patient should have received an initial considerably higher dose.
In recent years therapeutic drug monitoring (TDM) is recommended in mycobacterial disease, in particular for rifampicin. Please rephrase and add information on TDM and the importance of an adequate rifampicin dose and it´s dose dependent killing in mycobacterial disease both in the introduction and discussion.
Minor concerns:
3. The following sentence should be slightly revised since the majority of M. marinum cases are easily treated, "Still, the infections may lead to serious, treatment refractory skin and soft tissue lesions when that are diagnosed with a considerable delay
4. In figure 3 please consider a pre-and post treatment MRI (or normal MRI) including arrows pointing out abnormalities for further clarity
5. Please rephrase this sentence: "In conclusion, M. marinum may cause complicated infections often with significant morbidity". T
Reviewer 2 Report
This is a well written, and clearly illustrated case report of an M. marinum caused tenosynovitis. Such cases are rare, and difficult to diagnose, thus sharing the case is clearly justified.
In my opinion, this case should be of interest to clinicians as well as microbiologists and other readers. The quality of illustrations and case description is good and easy to follow. The authors demonstrates that the diagnosis was not easily made, and that the treatment was complex and required a multidisciplinary approach.
Reviewer 3 Report
Thanks for this manuscript, I greatly enjoyed reading it.
A couple of suggestions:
* The differential diagnosis of skin/soft tissue infections related to water exposure (AVEEM) could be described in more depth.
* Interestingly, IGRA was positive in this patient. I think this is an interesting finding and deserves some elaboration, e.g., Patel PM, Camps N, Rivera CI, Tuda C, VanOstran G. Mycobacterium marinum Infection and Interferon-Gamma Release Assays Cross-Reactivity: A Case Report. Cureus. 2022 Jan 19;14(1):e21420. doi: 10.7759/cureus.21420. PMID: 35198325; PMCID: PMC8856640.
